# Biogenic Fabrication of Iron Oxide Nanoparticles from *Leptolyngbya* sp. L-2 and Multiple In Vitro Pharmacogenetic Properties

**DOI:** 10.3390/toxics11070561

**Published:** 2023-06-27

**Authors:** Lubna Anjum Minhas, Muhammad Kaleem, Malik Abrar Hassan Minhas, Rooma Waqar, Dunia A. Al Farraj, Mona Abdullah Alsaigh, Hussain Badshah, Muhammad Haris, Abdul Samad Mumtaz

**Affiliations:** 1Department of Plant Sciences, Quaid-i-Azam University, Islamabad 45320, Pakistan; lubnaminhas@bs.qau.edu.pk (L.A.M.); mkaleem@bs.qau.edu.pk (M.K.); rooma.waqar667@gmail.com (R.W.); hbadshah@bs.qau.edu.pk (H.B.); m.haris@bs.qau.edu.pk (M.H.); 2Department of Chemical and Biological Engineering, University of Sheffield, Sheffield S10 2TN, UK; 3Department of Physics, Faculty of Basic and Applied Sciences, International Islamic University, Islamabad 44000, Pakistan; abrarhassan3@gmail.com; 4Department of Botany and Microbiology, College of Science, King Saud University, P.O. Box 24552, Riyadh 11451, Saudi Arabia; dfarraj@ksu.edu.sa; 5Department of Chemistry, College of Science, King Saud University, P.O. Box 24552, Riyadh 11495, Saudi Arabia; malsaigh@ksu.edu.sa

**Keywords:** iron oxide nanoparticles, *Leptolyngbya* sp. L-2, biocompatibility, antimicrobial, antileishmanial, enzyme inhibition

## Abstract

Metallic nanoparticles have received a significant amount of reflection over a period of time, attributed to their electronic, specific surface area, and surface atom properties. The biogenic synthesis of iron oxide nanoparticles (FeONPs) is demonstrated in this study. The green synthesis of metallic nanoparticles (NPs) is acquiring considerable attention due to its environmental and economic superiorities over other methods. *Leptolyngbya* sp. L-2 extract was employed as a reducing agent, and iron chloride hexahydrate (FeCl_3_·6H_2_O) was used as a substrate for the biogenic synthesis of FeONPs. Different spectral methods were used for the characterization of the biosynthesized FeONPs, ultraviolet-visible (UV-Vis) spectroscopy gave a surface plasmon resonance (SPR) peak of FeONPs at 300 nm; Fourier transform infrared (FTIR) spectral analysis was conducted to identify the functional groups responsible for both the stability and synthesis of FeONPs. The morphology of the FeONPs was investigated using scanning electron microscopy (SEM), which shows a nearly spherical shape, and an X-ray diffraction (XRD) study demonstrated their crystalline nature with a calculated crystallinity size of 23 nm. The zeta potential (ZP) and dynamic light scattering (DLS) measurements of FeONPs revealed values of −8.50 mV, suggesting appropriate physical stability. Comprehensive in-vitro pharmacogenetic properties revealed that FeONPs have significant therapeutic potential. FeONPs have been reported to have potential antibacterial and antifungal properties. Dose-dependent cytotoxic activity was shown against *Leishmania tropica* promastigotes (IC50: 10.73 µg/mL) and amastigotes (IC50: 16.98 µg/mL) using various concentrations of FeONPs. The cytotoxic potential was also investigated using brine shrimps, and their IC50 value was determined to be 34.19 µg/mL. FeONPs showed significant antioxidant results (DPPH: 54.7%, TRP: 49.2%, TAC: 44.5%), protein kinase (IC50: 96.23 µg/mL), and alpha amylase (IC50: 3745 µg/mL). The biosafety of FeONPs was validated by biocompatibility tests using macrophages (IC50: 918.1 µg/mL) and red blood cells (IC50: 2921 µg/mL). In conclusion, biogenic FeONPs have shown potential biomedical properties and should be the focus of more studies to increase their nano-pharmacological significance for biological applications.

## 1. Introduction

Nanotechnology (NT) is believed to be at the forefront of the development of nanomaterials, which are applied in various fields of science and technology [1]. NT involves the development of NPs that are helpful in the biomedical area, especially in drug delivery [2]. Fe is one of the most relevant infrastructures today due to its extensive usage in both biological and geological mechanisms [3]. FeONPs are the smallest iron metal particles, which are non-toxic, magnetic, and highly reactive, have a large surface area, strong thermal and electrical conductivity, and excellent dimensional stability. When exposed to water or air, FeONPs can instantly oxidize and release free iron ions. These nanoparticles have several uses, but the most fascinating is their function in drug delivery [4].

There are many conventional methods for the manufacture of FeONPs. Conventional techniques, such as physical and chemical, involve toxic and expensive chemicals and substantial use of energy. According to the literature, the chemical methods used to synthesize iron oxide nanoparticles include wet chemical, electro-deposition, chemical micro-emulsion, spray pyrolysis [5], microwave-assisted combustion, and chemical and direct precipitation. Physical methods include high vacuum usage in processes such as thermal evaporation, pulsed laser deposition, and molecular bean apitaxy [6]. To reduce the use of energy and chemicals, the biological methods are compatible, cost-effective, stable, and environmentally friendly [7]. Therefore, compared to chemical and physical synthesis methods, biological synthesis of FeONPs has more advantages in terms of high biocompatibility, non-toxic reagents, low energy requirements, and quick, easy-to-handle, easy-to-scale, and simple synthesis of nanoparticles. In other words, because they are accessible locally and affordable, plants have been employed as “biofactories” to produce metal nanoparticles. The synthesis of FeONPs uses significant phytochemical components (terpenoids, flavonoids, etc.) found in plant extracts with functional groups such as amine, hydroxyl, and carboxylic as reducing agents [8].

Fungi, bacteria, viruses, plants, and algae serve as reducing agents in biogenic synthesis [9,10]. Among many microbes, algae have demonstrated promising multiple biological potentials for NP synthesis, as well as the benefits of wastewater bioremediation, bioenergy generation, and large-scale production of commercial commodities such as pigments and pharmaceutical products [11]. Algae have a lot of functional groups, such as hydroxyl, carboxyl, and amino groups, which are involved in the synthesis of metal NPs [12]. These organisms have been studied as a modern alternative bio-agent for the biogenic synthesis of FeONPs [13,14]. The literature has described the use of algae in the manufacture of metal nanoparticles [14,15,16,17,18,19,20].

Cyanobacteria, also known as blue-green algae, are photosynthetic prokaryotes. They inhabit a wide range of environments and a variety of habitats [21]. Many cyanobacterial species have the unique capacity to fix atmospheric nitrogen (N2) by converting the gas to ammonia using nitrogen reductase enzymes in addition to oxygenic photosynthesis [22]. One of the most prevalent and taxonomically challenging cyanobacterial genera, Leptolyngbya is found in soils, periphyton, and metaphyton in a variety of freshwater and salty (marine) settings. It has a large number of morphotypes and ecotypes (species) [23]. They have various applications in agriculture, dietary supplements, bioremediation, and the biomedical industry as antibacterial and anticancer agents, making them a good source of secondary metabolites [24]. The advancements in nanotechnology provide new opportunities to generate new materials with unique antimicrobial capabilities [25]. The researchers have described various processes that contribute to FeONPs’ antibacterial potential. Among these processes, reactive oxygen species synthesis is the most crucial because it provides the antimicrobial potential for FeONPs. Furthermore, some other variables, such as surface defects in the symmetry of NPs, cause membrane injury due to nanoparticle adsorption on the surface, which causes cell lysis [26]. The ability to use biosynthesized nanoparticles in biomedical applications or directly in biological systems is the main advantage. This is due to the fact that these nanoparticles are less hazardous than those produced by physicochemical processes [27]. Numerous secondary metabolites, such as vitamins, phycobiliproteins, antioxidants, enzymes, and phenolic compounds, are frequently employed as anticancer, anti-HIV, antimalarial, antifungal, or antibacterial medications, are abundant in cyanobacteria [27]. According to reports, blue-green algae are a fantastic source of bio systems for the synthesis of NPs, either intracellularly or extracellularly. There are several publications in the literature that describe the use of blue-green algae for the biosynthesis of nanoparticles [28]. For instance, cyanobacteria *Leptolyngbya valderiana* [29], *Leptolyngbya* sp. WUC 59 [30], *Nostoc ellipsosporum* [31], *Anabaena flos-aquae* and *Calothrix pulvinata* [32] *Oscillatoria limnetica* [33] were employed for the biogenic synthesis of FeONPs. *Oscillatoria limnetica* and *Desertifilum* sp. have also been used in recent experiments to synthesize silver nanoparticles, which have demonstrated that the biological method is inexpensive and eco-friendly, and has a diverse array of biomedical applications [34,35].

To the best of our knowledge of the literature, this is the first report proving the potential use of *Leptolyngbya* sp. L-2 for the synthesis of FeONPs. Various approaches were used to further characterize the biogenic FeONPs. Moreover, a variety of bioactivities have been carried out, including antibacterial, antifungal, cytotoxicity, biocompatibility, alpha-amylase, and protein kinase inhibition assays. Through testing the synthesized nanoparticles and their efficiency, we allowed them to be able to be used in various fields without causing any harm.

## 2. Materials and Methods

### 2.1. Cyanobacterium

Cyanobacterium *Leptolyngbya* sp. L-2 was collected from Rawalpindi, Pakistan. The collected cyanobacterium was cultured using BG11 (blue-green algae media) media, and incubated in a growth chamber at 25 °C with a photoperiod of 8/16 h of dark under cool white light at 3000 Lux. *Leptolyngbya* sp. L-2 was identified based on morphological and molecular analysis. Morphological analyses were carried out using the light microscopic technique and molecular analyses were performed using NCBI BLAST searches and phylogenetic analysis through the Neighbor Joining Method using Mega 11.0.10 software.

### 2.2. Biogenic Synthesis of Targeted FeONPs

A total of 3 g of iron chloride hexahydrate (FeCl_3_·6H_2_O) was mixed with 50 mL of filtered *Leptolyngbya* sp. L-2 extract. To ensure a high reaction rate, the reaction mixture was heated on a hot plate for two hours at 70 °C while being constantly stirred. After cooling the reaction for two hours, the pH was found to be 9.73. Initially, a shift in color from light brown to dark brown served as a marker for the formation of IONPs. For the resulting solution, centrifugation was carried out at 3000 rpm for 30 min (Figure 1a). To get rid of iron ions and *Leptolyngbya* sp. L-2 extract residues, the pellet composed of FeONPs was washed three to four times with distilled water. The final powder designated as FeONPs was incubated at 75 °C for 2 h. The dried material was then reduced to fine powder using a pestle and mortar and calcined for 2 h at 600 °C to remove any impurities [27]. The calcination was performed in an open-air furnace. Later, the calcined FeONPs were maintained in a dry, cool, and dark environment and characterized using various methods. Figure 1b reveals the biological reduction of FeONPs.

### 2.3. Physical and Chemical Characterization of FeONPs

The biogenic FeONPs were characterized using a variety of methods, including UV, FTIR, XRD, SEM, EDX, and Zeta potential. The rate of metal ion reduction and biogenic synthesis of FeONPs were evaluated using an ultraviolet visible spectrum (UV-4000 spectrophotometer, Berlin, Germany). In the 200–600 nm region, aliquots of biosynthesized FeONPs were scanned while the background spectra were calibrated against DMSO. Biomolecules involved in the capping, reduction, and effective stability of FeONPs by the KBr pellet (0.2% to 1%) method were examined using the FTIR spectrum in the 500–4000 cm^−1^ region. This KBr technique takes advantage of the fact that alkali halides transform into plastic when put under pressure, forming a sheet that is transparent in the infrared spectrum. The most prevalent alkali halide utilized in the pellets is potassium bromide (KBr). The concentration of the sample in KBr should be in the range of 0.2 to 1 percent [36]. The software Origin Pro 8.5 was used to analyze the FTIR spectra.

X-ray diffractometer (PANalytical, The Netherlands, Europe) equipped with Cu radiation sources was used in the XRD investigation for FeONPs, and the Scherrer equation was used to calculate the mean crystalline size. Other components related to FeONPs were studied using EDX analysis. The EDX measurements show the atomic composition of nanoparticles and depict the atomic structure of NPs produced through biological methods. SEM was used to image and examine the shape and size of FeONPs. DLS used Zeta-sizer equipment to assess the zeta potential, polydispersity index (PDI), and hydrodynamic size distribution (Malvern Zetasizer Nano, Japan). For this, a bath sonicator was used to disperse 200 µg of FeONPs in 2 mL of deionized water (Milli-Q, Germany) for 40 min. After that, a foldable capillary cell was used to transfer the suspension for conductivity and zeta potential measurements.

### 2.4. Biomedical Properties of FeONPs

#### 2.4.1. Antibacterial Activity of FeONPs

Different bacterial strains were subjected to in-vitro testing for FeONPs’ antibacterial activity [37]. Prior to the antibacterial activities, the bacterial culture was revived, multiplied, and placed into test tubes containing nutrient broth media. The glass test tubes were then incubated at 200 rpm and 37 °C in a shaking incubator for 24 h. Cotton swabs were used to distribute the bacterial strains over medium plates. The antibacterial potentials were assessed using the disc diffusion technique. The filter disc (6 mm) was filled with 10 mL of each FeONP (with a concentration of 50–200 µg/mL), dried, and placed on bacterial lawns to prevent the spread of the test sample. An effective control was 10 mL of Oxytetracycline. After all, Petri plates were maintained at 37 °C in an incubator for 24 h, the emergence of a zone of inhibition (ZOI) was monitored at regular intervals, and minimum inhibitory concentrations (MIC) were recorded.

#### 2.4.2. Antifungal Assay

The antifungal activity of FeONPs was tested using the food poisoning method [37]. On PDA media, preserved cultures of fungi were cultivated for 72 h at 26.1 °C. Iron oxide nanoparticles in various doses (ranging from 50 to 200 µg/mL) were mixed with PDA media for this investigation. A cork borer was used to insert a 4 mm fungal inoculum disc into the center of the PDA plates. The Positive control was PDA media without any nanoparticles.

#### 2.4.3. Alpha Amylase Inhibition (α-AI)

The enzyme inhibition efficacy of *Leptolyngbya* sp. L-2 FeONPs was investigated to assess the potential for α-AI [38]. Alpha amylase, 40 mL of starch solution, 15 mL of phosphate-buffered saline, and 10 mL of the test sample were combined to produce the reaction mixture before being added stepwise to the microplate. The microplate was incubated for 30 min at 50 °C. Twenty minutes of incubation was followed by the addition of 20 mL of hydrochloric acid (1 M) and 90 mL of iodine solution to the mixture. As positive and negative controls, respectively, acarbose and DMSO were employed. The optical density (540 nm) was calculated using the microplate reader.

#### 2.4.4. Protein Kinase Inhibition (PKI)

The PKI potential of Streptomyces 85E was investigated using various concentrations of FeONPs [39]. The experiment was conducted in an aseptic environment. Streptomyces 85E strains were made for this purpose using ISP4 media. For making the uniform lawns, 100 mL of inoculum from the standardized culture was added and spread out uniformly throughout the Petri dishes. Six-mm sterilized filter discs containing 10 mL of FeONPs were carefully stored on microbial lawns. The positive and negative controls, respectively, were surfactin and DMSO. The Streptomyces 85E strain’s growth was targeted during the 72-h incubation at 30 °C on the Petri plates. Around the discs, various bald and clear zones developed after 72 h. These zones control spore inhibition and mycelia growth. The inhibition zone was determined in millimeters using a Vernier caliper.

#### 2.4.5. Brine Shrimps Cytotoxicity

The cytotoxicity potential of *Leptolyngbya* sp. L-2-mediated FeONPs was investigated against newly hatched Artemia salina [37]. Prior to the activity, Artemia salina eggs were incubated for about 24 h at 30 °C in artificial seawater (3.8 g/L) with light to develop mature nauplii. The test sample and 10 nauplii were placed in a glass vial with sea water using a Pasteur pipette. To investigate the cytotoxicity, various doses of FeONPs, ranging from 3.19 to 500 µg/mL, were employed. Positive control has consisted of a glass vials containing vincristine, mature nauplii, and seawater, whereas a negative control has consisted of glass vials containing seawater, DMSO, and mature nauplii. The vials were placed in an incubator for 24 h at 30 °C while the number of dead shrimps inside each was meticulously counted. The IC50 value for FeONPs was determined using the GraphPad program 8.0.

#### 2.4.6. Antileishmanial Assay

The cytotoxic ability of *Leptolyngbya* sp. L-2 mediated FeONPs were also tested against *L. tropica* (Promastigotes and Amastigotes cells) [38]. Promastigotes and amastigotes of both types of parasites were raised for this purpose in the MI99 medium with 10% FBS. To explore the antileishmanial potentials, *Leishmania tropica* was exposed to various concentrations ranging from 3.19 to 500 µg/mL. The reaction mixture was made up of 50 mL of fresh medium, 50 mL of colloidal NPs suspension in DMSO and 100 mL of the standardized culture. As positive and negative controls, respectively, amphotericin B and DMSO were utilized. The FeONPs test sample was placed in an incubator in a 96-well plate for 72 h at 24 °C with 5% CO_2_. A microplate reader was used to take readings at 540 nm after treatment and incubation with FeONPs. With the use of an inverted microscope live promastigote and amastigote were counted and IC50 values were recorded using the GraphPad program.

#### 2.4.7. Biocompatibility against Human RBCs

The biocompatibility experiment was carried out to validate the FeONPs’ biosafety when used with human red blood cells. For this objective, the anti-hemolytic assay was performed using human red blood cells [40]. For the biocompatibility test, 1 mL of freshly collected human RBCs was placed in an EDTA tube and centrifuged for ten minutes at 13,000 rpm. After that, the supernatant was discarded, and the pallet was thoroughly washed with phosphate buffered salines. 200 mL of erythrocytes were dissolved in 9.8 mL of PBS to prepare erythrocytes suspension. Different concentrations of FeONPs were mixed with 100 mL of the erythrocyte solution and placed in an incubator for 1 h, followed by centrifugation for 10 min at 10,000 rpm. The release of hemoglobin at 540 nm was measured using a microplate reader and a 96-well plate with the supernatant. Triton X-100 and DMSO were employed as the corresponding positive and negative controls.

#### 2.4.8. Biocompatibility against Human Macrophages (HMs)

Additionally, HMs were used to investigate the biocompatibility of FeONPs [41]. HMs were harvested using Ficoll GastroGrafins processes. This experiment was conducted gradually by combining 95 mL of distilled water, 5 mL of gastro-grains, and 5.7 g of fiber. After being diluted with Hank’s buffer salt solutions, the collected blood cells were put on Ficoll GastroGrafins and centrifuged for 30 min. Using a percoll gradient that had been adjusted with distilled water, the cells were cleaned. RPMI medium was used for cell culture. Additionally, cells were cultured in an incubator with 5% CO_2_ at a density of 1–105 cells per well.

#### 2.4.9. Antioxidant Assay

The effectiveness of FeONPs at scavenging free radicals was assessed using the spectrophotometric technique. To conduct this study, 2.4 mg of DPPH was carefully vortexed into 25 mL of methanol as a free radical. The ability to quench free radicals was then tested using different FeONP concentrations ranging from 3.19 to 500 µg/mL. Ascorbic acid was used as a positive control, and DMSO was used as a negative control. A 96-well plate was used to hold the 200 mL reaction mixture, which included 20 mL of FeONPs and 180 mL of reagent solution. A microplate reader was used to take measurements at 517 nm.

Additionally, the potassium ferricyanide procedure was used to assess the TRP (total reducing power) of FeONPs [42]. DMSO was used as the negative control, and ascorbic acid as the positive control. Using a microplate reader, the absorbance capacity at 630 nm was measured. The results of reducing power are expressed as mg of ascorbic acid equivalent per mg of NPs (mg AA/mg). The total antioxidant capacity (TAC) was calculated using the traditional approach, phosphomollybdenum [43]. At 695 nm, the absorbance was calculated using Eliza Reader. DMSO served as the negative control, and ascorbic acid as the positive control. The outcomes are shown as milligrams of ascorbic acid equivalent to each milligram of the test sample. The study scheme of the biosynthesis, characterization, and biological activities of iron oxide NPs shown in Figure 2.

### 2.5. Statistical Analysis

The results of the DPPH radical scavenging activity, total antioxidant activity, Cytotoxicity, antimicrobial, enzyme inhibition and biocompatibility assays of phyco-synthesized silver oxide nanoparticles are expressed as means ± standard deviations of the responses of three replicates per sample. Statistical analysis was performed with Microsoft Excel 365, Oringlab 2023 (10.0) and GraphPad Prism 8.0.

## 3. Results and Discussion

### 3.1. Physicochemical Properties of FeONPs

Cyanobacteria-mediated biogenic synthesis of FeONPs started after mixing the precursor salt iron chloride hexahydrate (FeCl_3_·6H_2_O) and *Leptolyngbya* sp. L-2 extract. The change in color of FeONPs/*Leptolyngbya* sp. L-2 solution at 60 °C from brown to dark black (Figure 3a) The color change indicates the presence of FeONPs. This chemical process is a result of the surface resonance phenomenon, with the reduction of Fe^+2^ to Fe^0^ and the formation of Fe_2_O_3_ [44]. The shift of the plasmon resonance during the coating process was quantified from the UV/vis absorbance spectra [45]. Surface plasmon resonance (SPR) is the manifestation of a resonance effect due to the interaction of conduction electrons in metal nanoparticles with incident photons [46]. The UV spectrum was used for further indication of the FeONPs’ synthesis in an aqueous environment. Figure 3c shows the UV spectrum of FeONPs, which were scanned between 200 and 600 nm. The highest absorbance peak demonstrated the conversion of salt into FeONPs within the range of SPR (318–608 nm). The stability of FeONPs was recorded at 300 nm. After 48 h, there is a reduction in the absorption peak, which is consistent with FeONPs’ settlement at the bottom of the falcon (Figure 3b). The size, shape, and stability of NPs are all clearly shown by UV-visible spectrum analysis [46].

XRD is a very useful technique to investigate the formation of crystals and to evaluate the crystalline size of the synthesized NPs. As demonstrated in Figure 4a,b, biogenic IONPs were annealed at 500 °C and then subjected to X-ray diffraction examination. The single, pure rhombohedral hematite phase of IONPs was found to match the observed Bragg peaks (JCPD card no. 00-005-0628). The diffractogram shows several diffraction peaks with 2 theta (2) values of 31.38, 34.03, 35.86, 47.18, 56.33, 62.48, 67.66, and 68.93, which are indexed to Bragg’s reflection at 200, 119, 209, 220, 024, 018, 214, and 300. Furthermore, using the high intensity peak in the (209) plane of the diffractogram, the Scherrer equation was used to calculate the crystalline size of the iron oxide nanoparticles of 28.21 nm each. Analysis revealed that the fabricated NPs were crystalline. These results agree with the iron oxide NPs previously synthesized by biological methods [47].

FTIR analysis was performed to confirm the presence of functional groups responsible for the biogenic synthesis and effective stability of FeONPs. The biomolecules present in the FeONPs were recorded at 500–4000 cm^−1^. *Leptolyngbya* sp. L-2 derived sulphated polysaccharides exhibited a significant capacity for NPs synthesis. Therefore, the presence of sulphate group may be responsible for the reduction of FeONPs by the oxidation of aldehyde groups to carboxylic acids [48]. The FTIR spectra of the *Leptolyngbya* sp. L-2 extract showed prominent absorption peaks of 3272, 2920, 1650, 1389, and 550 cm^−1^ (Figure 5). The presence of the hydroxyl group is detected by the peak at 3272 cm^−1^, amino groups at 1650 cm^−1^, and methylene groups at 2990 cm^−1^.These groups demonstrated the existence of aromatic rings and sulphated polysaccharides in *Leptolyngbya* sp. L-2. Therefore, *Leptolyngbya* sp. L-2 extract containing aldehyde, sulphate, and hydroxyl groups indicates the reduction of Fe^3+^ and biogenic synthesis of FeONPs [49].

SEM analysis (Figure 6a) was used to confirm the surface structure, particle size, and image size of FeONPs. The size of biogenic FeONPs ranged from 21 to 84 nm. In the SEM analysis, NPs had a polydisperse distribution and surface porosity that allowed the free movement of molecules [12]. EDX analysis determined the composition of the adsorbents of biosynthesized FeONPs (Figure 6b). The elemental composition and mapping of the components on the NPs surface were examined in the study. The prominent peaks represent the binding capacity of O and Fe atoms, which confirms the synthesis of FeONPs (Figure 6b) [50]. In EDX analysis, low peaks from elements such as C, N, K, and Ca are due to phyco-chemicals from cyanobacterium extract.

Zeta potential (ZP), analysis was used to confirm the surface charge of NPs that represents the electrical properties of sliding of NPs. The conditions for zeta potential measurements as shown in Table 1.

The electrostatic repulsion strength between adjacent and comparable charged particles in a dispersion is shown by the zeta potential degree. Nanosuspension with a ZP of +8.50 mV to −8.50 mV is considered a very stable form [51]. The zeta potential study revealed that the electrical charge of FeONPs was −8.50 mV (Figure 7). Figure 7a shows the intensity of scattering light on the y-axis and size on the x-axis. Zeta average size of FeONPs was 644.6 d·nm with PDI of 0.761 [4]. In algae based biogenic synthesis, ZP distribution was found to be negative (−) [52,53]. This negative value is due to the organic compound adhesion to the surface of the FeONPs [54]. The ZP measurements of *Leptolyngbya* sp. L-2 derived Biogenic FeONPs as shown in Table 2. ZP with a negative value exhibits agglomerated structures. In the present study, the negative zeta potential of *Leptolyngbya* sp. L-2 mediated FeONPs may be due to the chemical composition of the sealing and reducing agents used in biogenic synthesis as well as the NP’s particle size.

The comparative analysis of biogenic iron oxide nanoparticles (FeONPs) obtained from Leptolyngbya sp. L-2, in relation to relevant findings from previous studies, is presented in Table 3.

### 3.2. Pharmacogenetic Properties of FeONPs

#### 3.2.1. Antimicrobial Assay

The antifungal potential of FeONPs has studied against *Alternaria alternata* and *Botrytis cinerea* (Figure 8a). For positive control, Amphotericin B was used. A previous report has been conducted on the antibacterial potential of FeONPs while few antifungal tests have been recorded. This novel study is for the first to record the antifungal activity of *Leptolyngbya* sp. L-2 assisted FeONPs. Our results showed dose-dependent responses against these two strains. *Botrytis cinerea* was the least susceptible fungus strain, while the most susceptible strain was *Alternaria alternata*. The MIC was 50 µg/mL. At every concentration, the biogenic FeONPs demonstrated significant antifungal activity. However, no single concentration of ZOI outperformed Amphotericin B. Earlier research studies investigated the possibility that fungal growth may be stopped as a result of reactive oxygen species (ROS) and FeONPs with fungal hypea or spores. According to a previous study, FeONPs against different strains showed a dose-dependent response, which is consistent with the results of the current investigation [56].

The inhibitory potentials of FeONPs against different pathogenic bacterial strains are depicted in Figure 8b. The disc diffusion method was employed for this activity, with concentrations ranging from 200 to 50 µg/mL. *Staphylococcus aureus*, *Escherichia coli*, *Klebsiella pneumoniae*, and *coagulase negative Staphylococcus* were used for this study. Our findings showed that most bacterial strains had significant inhibitory potential and were sensitive to FeONPs. FeONPs were less effective at low concentrations (MIC: 50 µg/mL). Not a single dose was found to be more effective than that of oxytetracyclin. In summary, we have shown that FeONPs synthesized by biological methods exhibit strong antibacterial activity. The antibacterial potential of FeONPs has already been reported in earlier studies using various plants [57,58]. Furthermore, our research found that antibacterial potential increased with an increase in concentration. Additionally, various functional groups including phenolic acids, flavonoids, flavonols, and anthraquinonesa exist on the surface of FeONPs and come from *Leptolyngbya* sp. L-2 extract might be involved in the stability and capping of FeONPs, as a result, it plays a significant role in inhibiting various strains of pathogenic bacteria.

#### 3.2.2. Enzyme Inhibition Potential

The α-AI (alpha amylase inhibition) potential of FeONPs has been investigated. With a concentration range of 1200 to 37.5 µg/mL, the FeONPs have a strong ability to inhibit, with maximum potential for inhibition of 28% at 1200 µg/mL. α-AI functions by converting carbohydrates into glucose [59]. Alpha-amylase enzyme activity, and can lower blood sugar levels, which is an important field of diabetic research [60]. The FeONPs were studied for their α-AI potential. The findings indicate that FeONPs were found to be potent by inhibiting alpha amylase. When FeONP concentrations decrease, the rate of inhibition decreases. Surfactin (positive control) had a higher percentage of inhibition than any other concentration of FeONPs. The IC50 value was 3745 µg/mL and the maximum inhibition concentration was 28% at 1200 µg/mL. Figure 9a depicts the ability of FeONPs to inhibit α-A. Our results on biogenic FeONPs are consistent with previous studies [27,28]. Protein kinase enzymes play an important role in metabolism, cellular development, and apoptosis by adding phosphate groups to serine-threonine and tyrosine amino acids. Deregulated phosphorylation can cause genetic issues and cancer. As a result, any agent that can inhibit the protein kinase may be very important in cancer [59]. In the fungus *Streptomyces*, protein kinase phosphorylation is crucial for the development of hyphae, and similar mechanisms can be applied to investigate the potential of PKI (protein kinase inhibition). The strain was used as a model to validate the pharmacogenetic properties of compounds and calculate PKI [61]. Figure 9b shows that FeONPs inhibit the PK enzyme by a significant percentage. The disc diffusion technique was used to calculate the PKI potentials. Various concentrations of FeONPs were used to perform the PKI activity under sterile conditions. For positive control, surfactin was used. The zone of inhibition was 25 mm at a dose of 1200 µg/mL, which shows the significant PKI potential of FeONPs. FeONPs acted in a dose-dependent manner, according to our findings. The IC50 value was calculated as 96.23 µg/mL. Our findings on FeONPs are similar to earlier studies where green synthesized FeONPs showed potential results against PKI and can be further investigated for their potential role in the treatment of cancer [42].

#### 3.2.3. Cytotoxicity against Leishmania

Leishmania is a neglected tropical disease caused by *L. tropica* [62]. The parasites are endemic to 100 different countries, with an annual incidence rate of 1.2 million. The antileishmanial drugs conventionally utilized are often harmful, ineffective, and very costly. Antimonial was developed earlier as an effective candidate for the treatment of leishmaniasis, but the drug has lost its potential as *L. tropica* has developed resistance to it. As a result, the scientific community is working to create new methods of treating Leishmania. Since then, considerable research has been conducted on synthesized metal nanoparticles for the treatment of Leishmania. Several NPs have been used in in vitro studies to investigate cytotoxicity against leishmanial parasites [63]. However, little research has been conducted on the cytotoxicity of biogenic FeONPs against *L. tropica*. Figure 10a shows the antileishmanial activity of FeONPs. FeONPs were treated against Leishmania parasites at different concentrations ranging from 3.19 to 500 µg/mL for 72 h. In this study, *L. tropica* showed a dose dependent response [37]. As the concentration of FeONPs increases, % inhibition also increases. The IC50 value for FeONPs against *L. tropica* promastigotes has demonstrated promising antileishmanial potential (IC50; 10.73 µg/mL). Similarly, antileishmanial activity against *L. tropica* amastigotes was shown by FeONPs (IC50: 16.89 µg/mL), which is in agreement with earlier studies of biogenic FeONPs [64]. The dose-dependent and lower IC50 values of the FeONPs showed that FeONP materials can be used for strong antileishmanial drug delivery in future medicines.

#### 3.2.4. Cytotoxicity against Brine Shrimp

Brine shrimp cytotoxicity was analyzed to determine the cytotoxic potential of FeONPs against newly hatched *Artemia salina.* It is the best screening test to determine the biopotential of compounds [65]. The cytotoxicity of FeONPs was tested against *Artemia salina*. Figure 10b depicts the percentage mortality of FeONPs at different concentrations ranging from 3.19 to 500 µg/mL. Our findings revealed that cyanobacteria-mediated FeONPs showed a response in a dose-dependent manner. The IC5O value was determined as 34.19 µg/mL. Our results of *Leptolyngbya* sp. L-2 mediated FeONPs are comparable with previous reports of FeONPs [43].

#### 3.2.5. Biocompatibility Assays

It was determined whether FeONPs were biocompatible and toxicologically sound against human RBC and macrophages. The American Society’s fundamental principles state that biological compounds with less than 5% hemolysis are classified as non-hemolytic [66]. After receiving treatment with FeONPs, hemolysis is quantified by the rupture of the RBC, which releases hemoglobin. If the examined nanoparticles are hemolytic, the RBC will break, and heamoglobin will leak. As a result, the biocompatibility of IONPs was investigated using a hemolytic assay against human RBCs at concentrations ranging from 3.19 to 500 µg/mL. The current investigation has demonstrated that at concentrations below 31.25 µg/mL, the FeONPs are non-hemolytic, somewhat hemolytic at concentrations of 62.5 µg/mL, and hemolytic at concentrations above 62.5 µg/mL. The outcomes of *Leptolyngbya* sp. L-2 mediated FeONPs were found to be congruent with previous studies of FeONPs using *Rhamnus virgata* and *Sageretia thea* [67]. The IC50 value was calculated as 2921 µg/mL. Our study’s findings confirmed the biocompatibility and non-hazardous character of FeONPs by showing that biogenic FeONPs are non-hemolytic at low concentrations against RBCs. By conducting a cytotoxic experiment against human macrophages, the biosafety of FeONPs was further demonstrated. The study’s conclusion addressed the dose-dependent inhibitory response. When given higher concentrations of FeONPs, macrophages responded more favorably, whereas at lower concentrations, cytotoxicity decreased. The findings showed that FeONPs at 500 µg/mL suppressed macrophage development by about 24.8%, confirming their non-toxic nature. Typically, macrophages have acquired defenses against ROS (reactive oxygen species) generated from outside sources. Therefore, until its quantity exceeds the limit, ROS is not harmful to human macrophages and erythrocytes at low concentrations [68]. The FeONPs’ IC50 value was determined to be 918.1 µg/mL. The results of the biocompatibility experiments are shown in Figure 11a.

#### 3.2.6. Antioxidant Activities

Figure 11b depicts the antioxidant activities of FeONPs (TRP, TAC, and DPPH free radical scavenging). A range of 3.19 µg/mL was used for antioxidant tests. The highest result for total antioxidant in terms of AA per mg equivalents was determined to be 44.5% for FeONPs at 500 µg/mL. TAC revealed the scavenging potency of the tested nanoparticles to ROS species. Aqueous *Leptolyngbya* sp. L-2 extract was used in the current investigation as a reducing, capping, and oxidizing agent. Several phenolic compounds were inferred to scavenge the ROS, which is also bound to the FeONPs. TRP was examined to learn more about the existence of antioxidant species linked to FeONPs. This process was carried out to look at reductones that contribute H atoms to antioxidant potential, which may cause chain-damaging free radicals to develop [69]. The FeONPs described in this study have significant antioxidant potential. As the concentration of FeONPs decreased, so did their reducing power. The strongest reducing power (49.2%) was reported at 500 µg/mL. Strong DPPH radical scavenging capacity (54.7%) for FeONPs was seen at 500 µg/mL. According to the data in Figure 11b, a number of antioxidant chemicals may be involved in the decrease and stability of FeONPs in *Leptolyngbya* sp. L-2 extract. Our findings are consistent with past research on biogenic FeONPs [54,64].

## 4. Conclusions

In conclusion, our study has shown that a phytochemical-rich, toxin-free cyanobacterial *Leptolyngbya* sp. L-2 extract can be utilized to quickly and sustainably synthesize FeONPs. The formation of FeONPs can result from the interactions of several components in cyanobacterial extracts. The chemical and physical properties of biogenic FeONPs were successfully investigated. The reduction of iron metal ions to FeONPs was confirmed by the absorption band at wavelength 300 nm seen in UV-vis spectroscopy. The biogenic FeONPs nanoparticles had a size of around 28 nm. It is significant that this study also emphasized the numerous biological activities of FeONPs, showing that they have excellent antioxidant, biocompatible and enzyme-inhibiting properties. Our results showed that the potential of iron oxide nanoparticles showed dose-dependent behavior. This is the first investigation to highlight the biomedical properties of FeONPs using a blue green algal extract from *Leptolyngbya* sp. L-2. In the future, further in-vivo studies can be performed on the toxicity effect in various animal models, and if biocompatibility and biosafety can be ensured, then these biosynthesized FeONPs can be applied in different medical applications for the treatment of many chronic diseases.

## Figures and Tables

**Figure 1 toxics-11-00561-f001:**
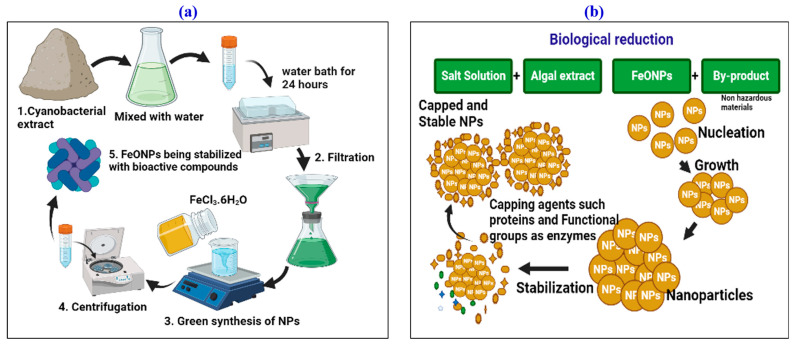
(**a**) Biological synthesis of FeONPs (**b)** Pictorial description of the possible mechanism for the biogenic synthesis of FeONPs and their effective stability.

**Figure 2 toxics-11-00561-f002:**
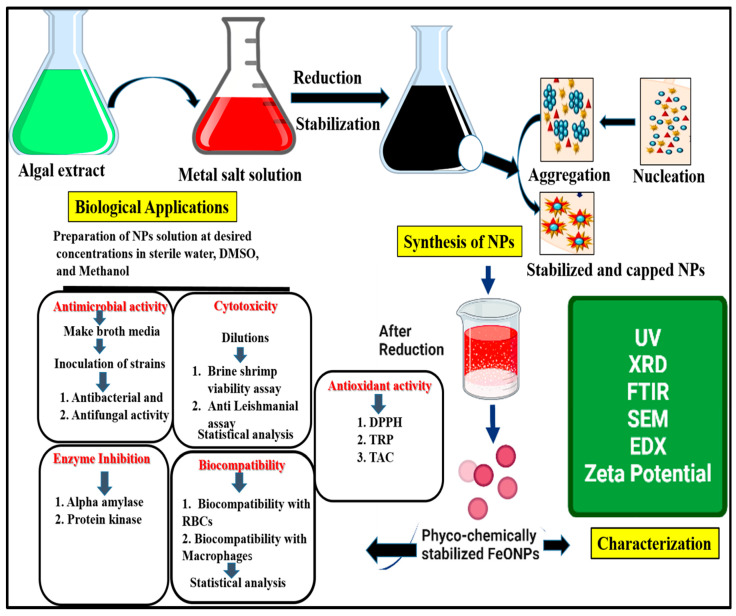
The mechanism of biogenic synthesis of *Leptolyngbya* sp. L-2 conjugated FeONPs.

**Figure 3 toxics-11-00561-f003:**
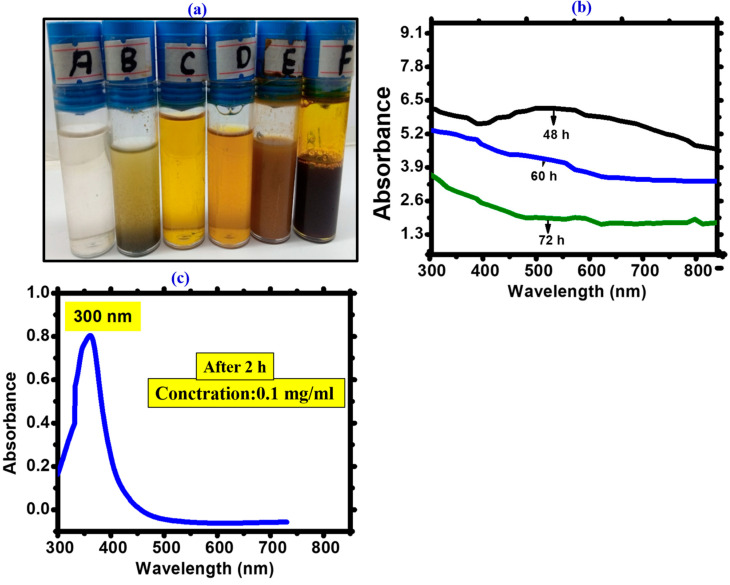
(**a**) Colour of FeONPs/ *Leptolyngbya* sp. L-2 solution at 60 °C (A) Distilled water (B) Extract of *Leptolyngbya* sp. L-2’ (C) Iron chloride hexahydrate (FeCl_3_·6H_2_O) salt solution (D) Mixture of extract and salt solution (pale yellow in appearance) (E) After one hour of treatment on hot plate it appears in light brown colour (F) After 24 h in open air it converted into iron oxide and it appear in dark brown colour. (**b**) Stability assessment depending on time measured after synthesis (after 24 h = 0.2 mg/mL, 48 h = 0.3 mg/mL and 72 h = 0.4 mg/mL) which shows particles settlement at the bottom of test tubes (**c**) UV visible spectrum of *Leptolyngbya* sp. L-2 derived Biogenic FeONPs.

**Figure 4 toxics-11-00561-f004:**
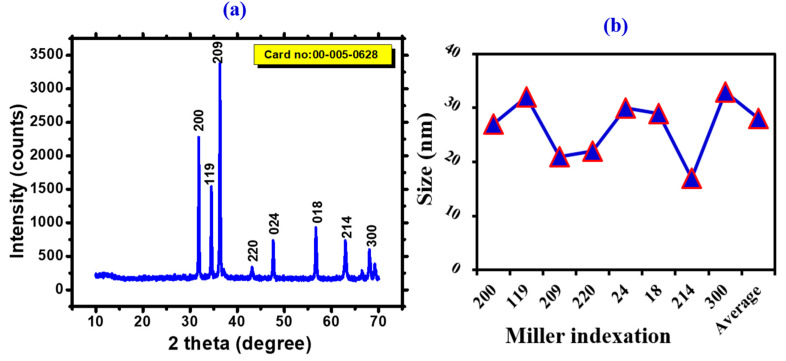
(**a**) XRD diffractogram of *Leptolyngbya* sp. L-2 derived Biogenic FeONPs (**b**) Size calculation via Scherrer equation.

**Figure 5 toxics-11-00561-f005:**
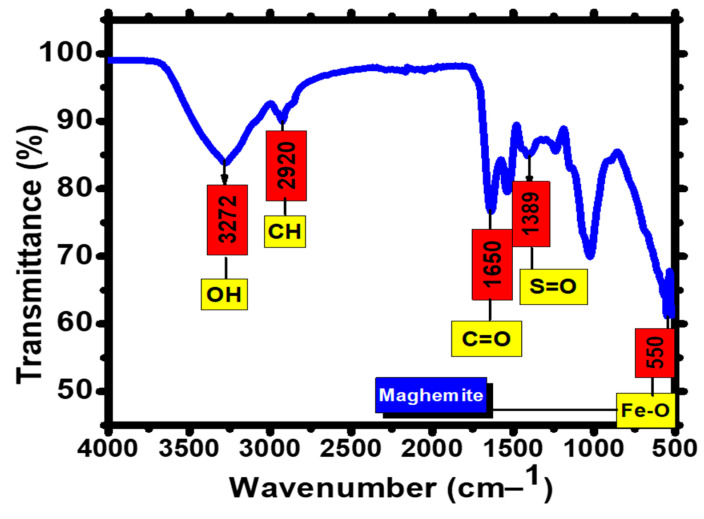
FTIR spectrum of *Leptolyngbya* sp. L-2 derived Biogenic FeONPs.

**Figure 6 toxics-11-00561-f006:**
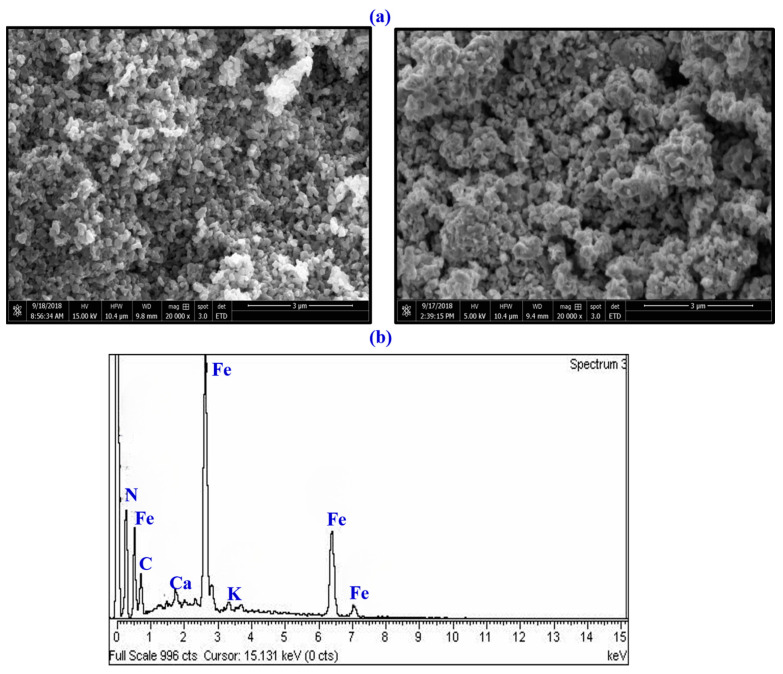
(**a**) SEM image of *Leptolyngbya* sp. L-2 derived Biogenic FeONPs (**b**) Elemental composition of *Leptolyngbya* sp. L-2 derived Biogenic FeONPs using EDX.

**Figure 7 toxics-11-00561-f007:**
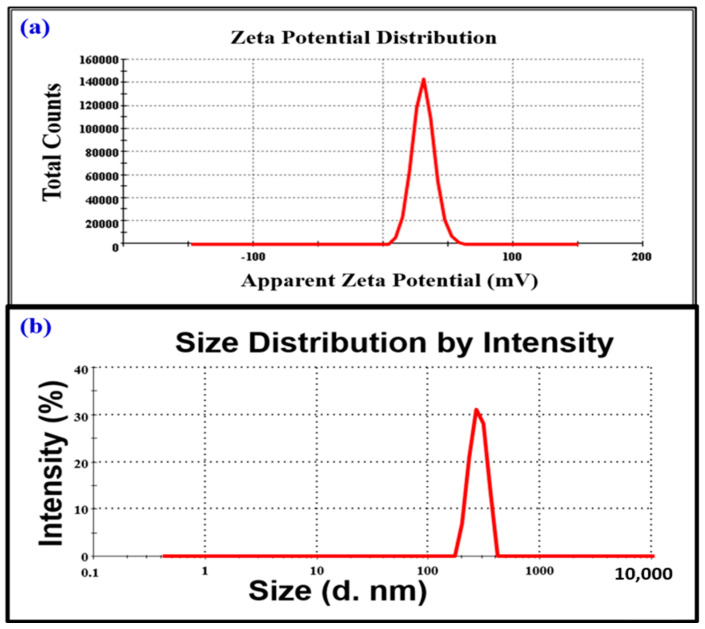
(**a**) Zeta Potential measurements of *Leptolyngbya* sp. L-2 derived Biogenic FeONPs, (**b**) Particle Size Distribution of *Leptolyngbya* sp. L-2 derived Biogenic FeONPs.

**Figure 8 toxics-11-00561-f008:**
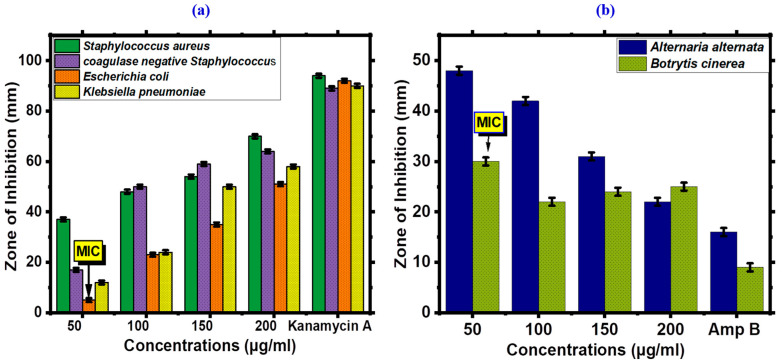
(**a**) Antifungal Potential of *Leptolyngbya* sp. L-2 derived Biogenic FeONPs, (**b**) Antibacterial Potential of *Leptolyngbya* sp. L-2 derived Biogenic FeONPs.

**Figure 9 toxics-11-00561-f009:**
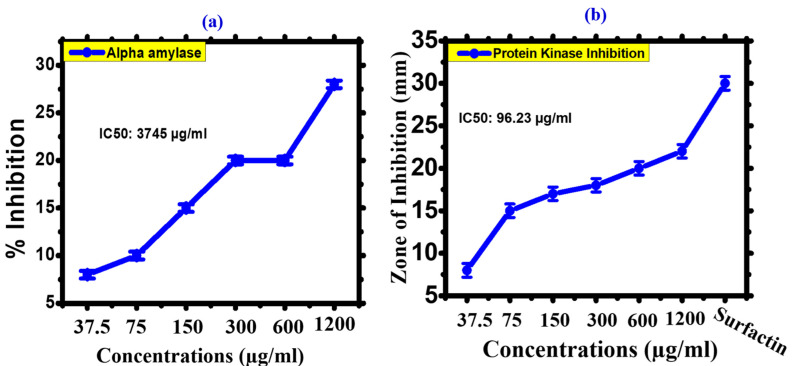
(**a**) Alpha amylase Inhibition of *Leptolyngbya* sp. L-2 derived Biogenic FeONPs, (**b**) Protein Kinase Inhibition of *Leptolyngbya* sp. L-2 derived Biogenic FeONPs.

**Figure 10 toxics-11-00561-f010:**
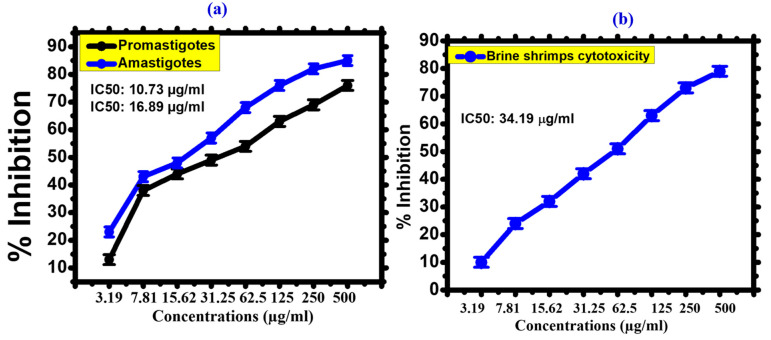
(**a**) Cytotoxicity activities of *Leptolyngbya* sp. L-2 derived Biogenic FeONPs against Leishmania, (**b**) Cytotoxicity activities of *Leptolyngbya* sp. L-2 derived Biogenic FeONPs against Brine Shrimps.

**Figure 11 toxics-11-00561-f011:**
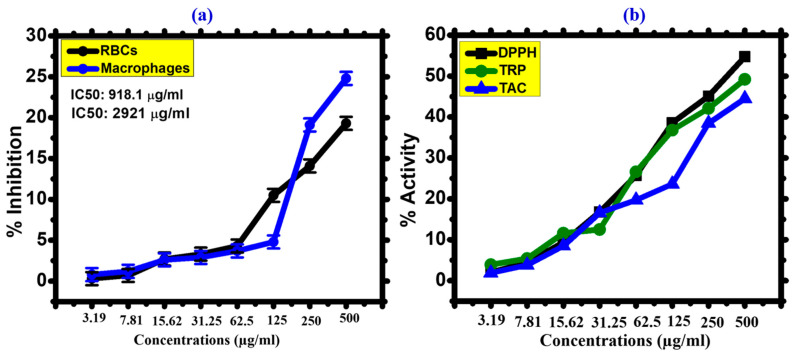
(**a**) Biocompatibility Potential of *Leptolyngbya* sp. L-2 derived Biogenic FeONPs against Human RBCs and Macrophages (**b**) Antioxidant Potential of *Leptolyngbya* sp. L-2 derived Biogenic FeONPs.

**Table 1 toxics-11-00561-t001:** Conditions for Zeta Potential (ZP) measurements.

Conditions for ZP Measurements	
Buffer Name	Phosphate-buffered saline
Dispersant RI	1.330
pH	7.4
Viscosity (cP)	0.8872
Dispersant Dielectric Constant	78.5
Temperature (°C)	25.0
Zeta Runs	12
Count Rate (kcps)	70.1
Measurement Position (mm)	4.50
Cell Description	Zeta dip cell
Attenuator	10

**Table 2 toxics-11-00561-t002:** Zeta potential measurements of *Leptolyngbya* sp. L-2 derived Biogenic FeONPs.

Zeta Size Distribution, Zeta Potential and PDI	Results
Zeta size	266
Zeta average size (d·nm)	644.6
PDI	0.761
Intercept	0.929
Zeta potential (mV)	−8.50
Zeta Deviation (mV)	16.1
Conductivity (mS/cm)	0.146
Result Quality	Good

**Table 3 toxics-11-00561-t003:** Comparative study of physical characterization of *Leptolyngbya* sp. L-2 derived Biogenic FeONPs.

Source	Method	Chemical	Size	Shape	Surface Morphology	Nature	Reference
*Oscillatoria* *limnetica*	Green synthesis	FeCl_3_·6H_2_O	23.33 nm	Trigonal rhombohedral	Agglomerated	Crystalline	[33]
*Spirulina platensis*	Green synthesis	FeCl_3_·6H_2_O	10 nm	Non-regular	Agglomerated	Crystalline	[55]
*Leptolyngbya* *valderiana*	Green synthesis	FeCl_3_·6H_2_O	47.42 nm	Spindle-like	Densely impregnated	Non-crystalline amorphous	[29]
*Chlorella* K01		FeCl_2_·4H_2_O	50 nm	Spherical	Monodispersed	Crystalline	[56]
*Leptolyngbya* sp. L-2	Green synthesis	FeCl_3_·6H_2_O	~28 nm	Spherical	Polydisperse	Crystalline	Current Work

## Data Availability

The data that support the findings of this study are available from the corresponding author upon request.

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
