# Peer review of "Biogenic Fabrication of Iron Oxide Nanoparticles from Leptolyngbya sp. L-2 and Multiple In Vitro Pharmacogenetic Properties"

_toxics, 2023, doi:10.3390/toxics11070561_

Round 1

Reviewer 1 Report

The manuscript “Biogenic Fabrication of Iron Oxide Nanoparticles from Leptolyngbya sp. L-2 and Multiple In-vitro Pharmacogenetic Properties” by Minhas et al. describes an investigation of iron oxide nanoparticles synthesized biogenically.

There are a lot of quality and reproducibility issues with this manuscript.

At the current stage, I cannot recommend to accept the manuscript.

Introduction

“There are many conventional methods for the manufacture of FeONPs. Conventional techniques such as physical and chemical involve toxic and expensive chemicals and substantial use of energy”

This statement is very vague. Can you please elaborate on synthetic processes and reference them in your introduction? It is not clear to which kind of syntheses you refer to, where toxic or expensive chemicals are needed and also where energy is needed.

Can you please describe the following method:

Biomolecules involved in the capping, reduction, 119 and effective stability of FeONPs by the KBr pellet method were examined using the FTIR spectrum 120 in the 500-4000 cm-1 region. The software Origin Pro 8.5 was used to analyze the FTIR spectra

How was the ratio of KBr to material for your spectra

Figure 2: The quality of figure 2 needs to be improved (looks elongated and small fonts are not readable).

Figure 3: (B) Stability of green synthesized FeONPs (C) UV visible spectrum of 248 Leptolyngbya sp. L-2 derived Biogenic FeONPs.

It is unclear how the UV/Vis spectra relate to the stability. Can you elaborate this in the caption? Especially, the concentration is missing.

Also the concentration for figure  3C is missing. I am also wondering why the intensity of the absorbance is decreasing with lower wavelength? Usually, all iron oxides (magnetite, maghemite, hematite and even ferrihydrites) show an increasing absorbance at these lower wavelengths. Is it possible that there is some cut-off in the spectrometer above 200 nm?

Line 254: The spectra XRD -> you have not analyzed a spectrum but a diffractogram. In a diffractogram different angles of one wavelength are investigated while in a spectrum you look at different wavelengths.

Line 257: the Debye-Scherrer equation was used to calculate the crystalline size

In this case it is just the Scherrer equation

Figure 4: This figure also looks elongated and should be changed.

The following caption “(A) XRD spectrum of Leptolyngbya sp. L-2 derived Biogenic FeONPs (B) Size calculation 262 via Scherer’s equation.” Contains two errors. First, a diffractogram and not a spectrum is shown and it is the Scherrer and not the Scherer equation.

Figure 5: The vibration at 1022 is likely C-O-C which corresponds to saccherides. However, I doubt that this is an S=O vibration.

Figure 6: The names of the elements should be increased in size sinze it is challenging to read them. Moreover, the quality of this figure can also be improved.

For the zeta potential study it is completely unclear unde which conditions this measurement has been conducted. Can you pleas give the buffer conditions as well as the pH.

The following sentence is misleading: “Nano emulsion with ZP of +8.50 mV to −8.50 mV is considered a very stable form [36].” First, you have a nanosuspension and not an emulsion. Second, a zeta potential higher 30 mV or lower than -30 mV is generally considered stable for nanosuspensions. Zeta potentials close to the isoelectric point are usually less stable and tend to agglomerate.

Your arguments considering negative charge due to biologic materials are void without considering the buffer and the pH conditions.

Figure 7: Please explain the conditions for the measurements.

Section 3.2.1 is basically a description of the state of the art and would fit better in the introduction than in the results.

Figure 9 has a low quality

Author Response

Dear Reviewer

Thank you

Reviewer 2 Report

The presented paper entitled “Biogenic Fabrication of Iron Oxide Nanoparticles from Lep- 2 tolyngbya sp. L-2 and Multiple In-vitro Pharmacogenetic Properties” is interesting and might advance the scientific community. However, several major concerns should be addressed before this paper can be accepted for publication. The authors should seriously consider the following major revisions:

1.     The abstract should be modified to better summarize the current work. For instance, Lines 26-28 should be re-phrased to give a powerful introduction to the topic.

2.     The introduction section should be extensively re-written, developed and broadened further, and more state-of-the-art studies similar to this work should be added.

3.     In the introduction, line 80 is not accurate because several studies are found in the literature reporting the biogenic synthesis if Iron NPs.

4.     The name of the involved algae should be fully stated in the title, abstract and introduction.

5.     In the introduction,the authors should elaborate more on Cyanobacterium Leptolyngbya alge in specific considering its origin, properties, and previous studies that used same species in green synthesis of NPs. For instance, https://doi.org/10.1016/j.algal.2021.102373

6.     In section 2.2, it is not clear how the green synthesis was conducted? Is it by mixing only.

7.     In section 2.2., the authors heated the mixture at 70 oC, for two hours. I believe this high temperature for 2 h would affect the ability of Leptolyngbya to reduce the iron. This is a serious issue that should be addressed very clearly.

8.     Also, in section 2.2., a reference supporting the protocol of preparation should be included.

9.     How did the authors claim that the Cyanobacterium Leptolyngbya extract has reducing ability and is responsible for the reduction of iron? Antioxidant activities of the extract should be tested using DPPH, and other techniques.

10.  Figure 1 should present the detailed steps involved in the biogenic synthesis. I do not think it involves only simple mixing.

11.  In the legend of Figure 3, (A) should be rephrased as it is very confusing.

12.  In line 296, the authors mentioned that “is due to the organic compounds adhesion to the surface of the FeONPs”. Thus tye following should be conducted:

a.      In Figure 3C, the authors should present the spectrum of the extract in comparison to the NPs, to compare the spectra of both, and determine if the extract influences the spectrum of the  NPs.

b.     Same should be done in the FTIR to compare and confirm if organic compounds from the extract are found on the surface of the NPs.

13.  Based on point 12, The synthesized nanoparticles retained a significant proportion of the organic fraction. The content of organic material in the nanoparticles should be determined, calculated and interpreted by thermogravimetric analysis and citing this paper https://doi.org/10.3390/nano11040965. These data are required to interpret the results of biological assays.

This is very important to be done.

14.  A statistical analysis sub-section should be introduced in the materials and methods section describing the statistical methods used in this work and the probability at which the results were considered significant.

15.  All biological assays should involve a positive controls (Antibacterial and antifungal, etc.). Also, the algal extract should be involved in all the biological assays as compared to the biogenic NPs, in order to make sure that the activities are attributed to the iron oxide NPs not the extract.

16.  The results and discussion section should be better presented to highlight the most significant and unexpected results, and identify correlations, patterns, and relationships among the data, speculations, limitations of work, and deductive arguments.Also, all results should be more integrated with the discussion and should be supported by state-of-the-art studies.

17.  The conclusions section should be improved. Ideally, the Conclusions section reviews the key findings of your work and explains the specific ways in which this work fundamentally advances the field relative to prior literature. The Conclusion section should summarize the manuscript's results and their importance, discuss ambiguous data and recommend further research. Furthermore, an effective conclusion should offer closure to a paper, leaving the reader feeling satisfied that the concepts have been fully elucidated.

Moderate revisions.

Author Response

Dear Reviewer

Thank you

Round 2

Reviewer 1 Report

The manuscript has been improved.

Reviewer 2 Report

The authors addressed most of the raised concerns. I endorse publication in the present form.

Moderate edits